# Snapshot of *Mycobacterium tuberculosis* Phylogenetics from an Indian State of Arunachal Pradesh Bordering China

**DOI:** 10.3390/genes13020263

**Published:** 2022-01-29

**Authors:** Shiv kumar Rashmi Mudliar, Umay Kulsum, Syed Beenish Rufai, Mika Umpo, Moi Nyori, Sarman Singh

**Affiliations:** 1Department of Microbiology, All India Institute of Medical Sciences, Bhopal 462020, Madhya Pradesh, India; srashmim1@gmail.com (S.k.R.M.); kulsum.umay@gmail.com (U.K.); 2Infectious Diseases and Immunity in Global Health Program, Research Institute of the McGill University Health Center, Montreal, QC H4A 3J1, Canada; beenish.aiims@gmail.com; 3McGill International TB Center, Montreal, QC H4A 3J1, Canada; 4Tomo Riba Institute of Health & Medical Sciences, Naharlagun 791110, Arunachal Pradesh, India; athupopu@gmail.com; 5State TB Cell, Naharlagun 791110, Arunachal Pradesh, India; stoar@rntcp.org

**Keywords:** DST, MGIT 960, WGS, cgMLST, SNP barcoding, phylogeny

## Abstract

Uncontrolled transmission of *Mycobacterium tuberculosis* (*M. tuberculosis*, MTB) drug resistant strains is a challenge to control efforts of the global tuberculosis program. Due to increasing multi-drug resistant (MDR) cases in Arunachal Pradesh, a northeastern state of India, the tracking and tracing of these resistant MTB strains is crucial for infection control and spread of drug resistance. This study aims to correlate the phenotypic DST, genomic DST (gDST) and phylogenetic analysis of MDR-MTB strains in the region. Of the total 200 samples 22 (11%) patients suspected of MDR-TB and 160 (80%) previously treated MDR-TB cases, 125 (62.5%) were identified as MTB. MGIT-960 SIRE DST detected 71/125 (56.8%) isolates as MDR/RR-MTB of which 22 (30.9%) were detected resistant to second-line drugs. Whole-genome sequencing of 65 isolates and their gDST found Ser315Thr mutation in *katG* (35/45; 77.8%) and *Ser*531Leu mutation in *rpoB* (21/41; 51.2%) associated with drug resistance. SNP barcoding categorized the dataset with Lineage2 (41; 63.1%) being predominant followed by Lineage3 (10; 15.4%), Lineage1 (8; 12.3%) and Lineage4 (6; 9.2%) respectively. Phylogenetic assignment by cgMLST gave insights of two Beijing sub-lineages viz; 2.2.1 (SNP difference < 19) and 2.2.1.2 (SNP difference < 9) associated with recent ongoing transmission in Arunachal Pradesh. This study provides insights in identifying two virulent Beijing sub-lineages (sub-lineage 2.2.1 and 2.2.1.2) with ongoing transmission of TB drug resistance in Arunachal Pradesh.

## 1. Introduction

India is leading in the highest rates of tuberculosis (TB) incidence and mortality globally, with an estimate of 2.69 million cases [1,2]. Although drug-resistant tuberculosis (DR-TB) is a major public health concern globally, it represents an alarming situation in India, with 135,000 MDR-TB cases contributing to 27% of global DR-TB cases [1]. Patients with DR-TB often require profound changes in their drug regimens, which are invariably linked to poor treatment adherence and sub-optimal treatment outcomes compared to drug-sensitive TB. Higher drug-resistant TB cases remain a challenge for clinicians and National Tuberculosis Elimination Programme (NTEP) for accurate and effective TB treatment in India [3,4]. In India, the paucity of rapid diagnosis in locations having low resources and high endemicity areas where access to health care centers is difficult, remains a major constraint in treating DR-TB cases. It is estimated that around 56% of MDR-TB cases in India remain undiagnosed [4]. Arunachal Pradesh, one of the states in the northeastern region of India bordering China with 80% area covered with forest, mostly with hilly terrains, has awakened consciousness of NTEPs due to the high prevalence of around 78.8% MDR-TB cases [5]. The reason for undiagnosed drug-resistant cases is the location of most villages in impoverished forest zones, poor connectivity of roads to health centers leading to inadequate access to health services.

There is increasing evidence that the inter-strain variation in *M. tuberculosis* exists due to variation in gene expression profiles and is biologically significant [6]. Several molecular epidemiological studies have proposed that certain types of *M. tuberculosis* strains that are prone to drug resistance may have rapid transmission rates and a higher rate of recurrence due to relapse [7]. Understanding the role of strain variation of *M. tuberculosis* to clinical phenotypes requires an approach to categorize *M. tuberculosis* isolates into groups that share most of the genotypic and phenotypic traits. For this, studies based on phylogenetic analysis, which organize clinical isolates into genetically related groups, are needed. Such studies provide an evolutionary framework for investigating polymorphisms and their potential biological relevance [8].

Molecular strain typing using the 24-loci MIRU-VNTR along with Spoligotyping is widely used for characterizing ongoing transmission of *M. tuberculosis* strains in a particular geographical location and has been shown to provide crucial information effective for the public health interventions [9]. However, these typing methods are reported to miss considerable amounts of genetic diversity, and where the overall diversity of circulating clones is limited, these approaches are insufficient to differentiate among strains [10].

With the advancement of Next-generation sequencing, Whole-genome sequencing (WGS) data are appraised for its use in epidemiological studies, strain typing for outbreak investigations, and surveillance of infectious disease due to higher sensitivity and rapid turnaround time [11]. Different genotyping methods such as spoligotyping and MIRU-VNTR were previously used for the categorization of lineages and sub-lineages. However, comparative analysis led to the use of single nucleotide polymorphisms (SNPs), which provided valuable insights into the epidemiology of circulating strains and were used as robust genetic signatures for phylogenetic categorization [12,13,14]. A well-established SNP barcode approach is already well known for analyzing 60 loci in *M. tuberculosis sensu stricto* genomes and has efficiently been used for categorization of major Lineages 1–7 and sub-lineages [13]. Moreover, core genome multilocus sequence typing (cgMLST) based on entire allele change is widely being used with confidence for assessing phylogenetic position to genomic data sets within a single species The sum of all core genes and their alleles for a species comprise the species cgMLST schema [14]. Phylogenetic heterogeneity, drug-resistant patterns, and association of lineages with drug resistance are not well known from the Arunachal Pradesh region of India. This study was aimed to utilize approaches of cgMLST and SNP barcoding to envision circulation of *M. tuberculosis sensu stricto* lineages. We also aim to perform phenotypic DST and genomic DST (gDST) to see the patterns of drug resistance in clinical strains of *M. tuberculosis*.

This study will help in the identification of hypervirulent strains/clones that are circulating among drug-resistant TB cases in Arunachal Pradesh and will provide crucial information to NTEP and public health programmers. Fast and accurate tracking of hypervirulent *M. tuberculosis* strains is, therefore, essential to keep track of ongoing circulating clones, which is decisive for infection control and can help in the prediction of potential future outbreaks.

## 2. Materials and Methods

### 2.1. Study Setting and Sample Collection

A total of 200 sputum samples (1 sample per patient) 22 (11%) patients suspected of MDR-TB, and 160 (80%) previously treated MDR-TB cases were collected from 6 districts of Arunachal Pradesh (Papum Pare, East Kameng, Kurung Kumen, Tirap, Lower Dibang Valley and Kra Daadi) by Department of Microbiology, Tomo Riba Institute of Health and Medical Sciences from September 2019 to September 2021. Samples were transported in triple package cold chain within 72 h of collection to TB laboratory, Department of Microbiology, AIIMS Bhopal for liquid culture and DST. Informed consent was collected from all study participants. Ethical clearance was obtained for carrying out the study at TRIHMS Arunachal Pradesh and AIIMS Bhopal under reference number DME (T&R)/IEC/2015/1 and IHEC-LOP/2018/EF0104, respectively.

### 2.2. Decontamination of Samples and Bactec MGIT 960 Culture Inoculation

Sputum samples were processed using NALC-NaOH method [15]. Briefly, a minimum 3 mL of sputum sample was mixed with an equal amount of 0.5% NALC-4% NaOH, vortexed and incubated at 37 °C for 10 min. Samples were then neutralized and washed with phosphate buffer (PH 6.8) by centrifuging at 10,000 rpm for 10 min. Pellet was resuspended in 2 mL of phosphate buffer and mixed well. Smears were prepared for Ziehl-Neelsen (ZN) staining.

Five hundred microliter of the decontaminated sample was inoculated in Bactec MGIT 960 culture tubes containing 800 µL mixture of oleic acid, albumin, dextrose, and catalase (OADC) and polymyxin B, amphotericin B, nalidixic acid, trimethoprim, azlocillin (PANTA) supplement as per the manufacturer’s instructions (Becton Dickinson Diagnostic Instrument Systems, Sparks, MD, USA). Leftover decontaminated sample aliquots were stored at −80 °C for future use.

### 2.3. Identification of Cultures Using In-House Multiplex PCR

Bactec MGIT 960 cultures were identified as *M. tuberculosis* complex using in-house multiplex PCR, which targets *hsp-65* (genus-specific), *esat-6* (MTB specific), and internal transcribed spacer (ITS) MAC region (MAC specific) [16]. Amplified products were resolved through 2% agarose gel in Tris–acetate buffer [16].

Identified cultures of *M. tuberculosis* complex were subcultured on slants of Lowenstein Jensen (LJ) medium and incubated at 37 °C for 21–28 days. Growth from LJ medium was used for Bactec MGIT 960 DST and DNA extraction for WGS [17].

### 2.4. Bactec MGIT 960 SIRE DST

Single colony with the help of sterile inoculating loop from each LJ medium was inoculated in each MGIT 960 system tube and incubated in Bactec instrument until flagged positive. First-line SIRE DST was performed as per the manufacturer’s protocol [18]. DST was performed on Day 1 and Day 2 by single dilution (0.5 mL of 1:100 dilution inoculum for growth control (GC) and 0.5 mL of inoculums directly in four drug panel tubes) and Day 3 to Day 5 (0.5 mL of 1:4 mL dilution inoculums directly for four drug panels and further dilution in 1:100 for GC) from the day of flagged positive of MGIT 960 instrument tube [18,19]. *M. tuberculosis* H37Rv [ATCC (American Type Culture Collection) number 2799] and known MDR-TB strain were used as quality control. The inoculated MGIT tubes with DST racks were loaded in the automated Bactec MGIT 960 system and the growth was continuously monitored by BD Epi-center.

### 2.5. Second Line DST Using Bactec MGIT 960

Isolates identified as MDR-TB were tested against second line drugs viz., moxifloxacin (MOX), levofloxacin (LEV), amikacin (AMK), and linezolid (LNZ). All drugs were purchased from Sigma-Aldrich Corporation (St. Louis MO, USA) and were chemically in the form of powder. Stock solution of drugs AMK (1 mg/mL), MFX (1 mg/mL), LFX (1 mg/mL) and LNZ (1 mg/mL) were prepared as per the instruction and sterilized through 0.22 μm pore-size Millex-GS filter units (Millipore Bedford, MA, USA). *M. tuberculosis* H37Rv [ATCC (American Type Culture Collection) number 2799] and known fluoroquinolone (FQ) resistant strains were used as quality control strains. Second-line DST was performed as per the protocol [19,20,21]. As AST carrier rack for second-line drug (SLDs) panels were not available commercially for the MGIT-960 system, it was registered as one of the SIRE (Streptomycin, Isoniazid, Rifampicin, Ethambutol) panel in order to obtain a printable report and drug susceptibility testing results.

### 2.6. Whole Genome Sequencing

Genomic DNA extraction from 65 clinical isolates (representatives of sensitive and resistant groups as described in Figure 1) of *M. tuberculosis* complex was carried out using the Chloroform-Isoamyl alcohol (CI) method [22], and quantification of genomic DNA was performed using Qubit Fluorometer (Thermo Fisher). The genomic DNA was sequenced using plexWell WGS-24 Library Preparation kit (Illumina, San Diego, CA, USA). Library pools were subjected to paired-end sequencing on a HiSeq platform (Illumina, San Diego, CA, USA).

The quality of sequenced reads was screened using FastQC v0.11.9, and the reads with an average quality score of >=20 were retained [23]. Reads that were shorter than 36bp and possible adapter contaminating sequences were removed using Trim Galore Version 0.6.4 [24]. The output of contigs/genomes were assembled using the SPAdes genome assembler (version 3.9.0) using the default k-mer size [25] and annotated using Prokaryotic genome annotation pipeline (PGAP) [26].

### 2.7. Identification of SNPs

Raw reads of each genome were mapped to *M. tuberculosis* H37Rv reference genome (Accession NC_000962.3) using Burrows–Wheeler Aligner (BWA-MEM algorithm) bwa-0.7.12 [27]. SAM to BAM format conversion and sorting of mapped sequences, filling in mate coordinates to keep best reads using mate score tags was performed using Samtools 1.10 [28]. Duplicate alignments were marked and removed using samtools markdup command. BAM files were indexed for piling up variants by samtools mpileup. Mutations with read depth above 10 reads were considered true mutations. Annotation and filtering of variants were conducted using SNPEFF 5.0e [29] and SnpSift [30].

### 2.8. Assignment of Principal Genetic Groups

To assign a principal genetic group (PGG), each sequenced isolate was manually screened for polymorphisms in *gyrA* codon 95 and *KatG* codon 463 and were categorized accordingly to PGG as 1, 2, or 3, respectively, as explained previously [31].

### 2.9. Identification of Lineages and Sub-Lineages Using WGS SNP Barcoding

Isolates based on the patterns of SNP at the designated loci were categorized into phylogenetic lineages groups as lineage 1 (Indo-Oceanic), lineage 2 (East Asian), lineage 3 (East African Indian), lineage 4 (Euro-American), lineage 5 (West Africa 1), lineage 6 (West Africa 2), and lineage 7 (Horn of Africa) and lineage 8 [13]. After splitting WGS isolates into lineages, further categorization was conducted on the basis of SNP’s [13]. Construction of the UPGMA tree was conducted by concatenating SNPs and visualized using iToL V.6 software [32].

### 2.10. Phylogenetic Analysis and Construction of cgMLST

Schema for cgMLST was set up with an efficient Workflow for a Blast Score Ratio Based Allele Calling Algorithm (Chew BBACA) [33]. For construction of cgMLST, first a wgMLST schema was created using *M. tuberculosis* H37Rv (accession: NC_000962.3) as a training file generated by the prodigal algorithm. The wgMLST schema contained 10058 loci based on 104 genomes (65 genomes sequenced under this study, 39 complete and draft genomes downloaded from NCBI). Complete genome sequences of *M. tuberculosis* complex viz; *M. bovis* (NC_002945.4), *M. orygis* (CP063804.1), *M. africanum* (FR878060.1), *M. cannettii* (NC_015848.1), *M. tuberculosis* (NC_000962.3) and *M. caprae* (NZ_CDHG01000001.1) were extracted from NCBI. Draft genomes from NCBI database that belong to Lineage 1 (PRJNA235648, PRJNA223559, PRJNA229273, PRJNA229212, PRJNA229320, PRJNA229266), Lineage 2 (PRJNA219760, PRJNA226779, PRJNA267047, CCDC5180, CCDC5079, PRJNA229630, NDYV00000000, PRJNA360122), Lineage 3 (PRJNA229310, PRJNA229235, PRJNA229259, NDYU00000000), Lineage 4 (PRJNA229257, PRJNA223558, PRJNA229237, PRJNA228063, PRJNA228052, PRJNA229638, PRJNA218312, PRJNA233359, PRJNA233363), Lineage 5 (PRJNA211660), Lineage 6 (PRJNA211707, PRJNA211702), Lineage 7 (PRJEB8432) and Lineage 8 (PRJNA598991) and lineage B (PRJNA229213), were extracted and used as representative for *M. tuberculosis* lineage 1-8. These 39 publicly available complete genomes were used for validation of the cgMLST schema. The resulting loci was then subjected to *AlleleCall*, which identified and excluded 104 possible paralogous loci from further downstream analysis using the default BLAST Score Ratio (BSR) threshold of 0.6. Finally, cgMLST was extracted containing a set of 1443 core loci (present in 100% of the isolates). The resulting cgMLST matrix was uploaded in phyloviz 2.0 [34] to generate and visualize UPGMA Tree. The genome of *M. cannettii* was used to root the tree.

### 2.11. Data Analysis

All data obtained from gDST, phenotypic DST, PGG, and categorization of lineage on the basis of SNP barcoding were maintained on MS Excel 2013 for further analysis.

## 3. Results

### 3.1. Demographic Details and Characteristics of MDR-TB Patients

Of the total 200 patients included in the study, 91 (45.5%) were males and 109 females with mean age (± standard deviation) of 29.52 ± 13.21 and 27.85 ± 14.17 years, respectively. The majority of cases were adults, 183 (91.5%), and 17 (8.5%) were from the pediatric age group.

### 3.2. Bactec MGIT 960 Culture Results and Identification *M. tuberculosis* Complex Isolates

One hundred and twenty six of 200 (63%) samples were smear-positive, and 74 (37%) smear-negative. Of the total 200 cultures inoculated in Bactec MGIT 960 145 (72.5%) were flagged positive with an average turnaround time (TAT) of 18 days. All flagged positive cultures were further confirmed by ZN-stained smear examination, of which 131/145 (90.3%) were smear-positive for AFB while 14/145 (9.7%) were contaminated. Of 131 cultures, in-house multiplex PCR identified 6 (4.6%) cultures as Non-tuberculosis Mycobacterium and 125 (95.4%) cultures as *M. tuberculosis* complex (Figure 1).

### 3.3. Bactec MGIT 960 SIRE DST

Of total 125 cultures, Bactec MGIT 960 SIRE DST detected 66 (52.8%) as MDR-TB (resistant to both RIF and INH), 5 (4.0%) as mono-resistant to RIF, and 14 (11.2%) were drug-resistant isolates (resistant to any of first-line drug other than MDR/RR). 40 (32.0%) isolates were detected as pan-sensitive (Table 1).

### 3.4. Bactec MGIT 960 Second Line DST

Of 71 (56.8%) MDR-TB and RIF mono-resistant isolates subjected to second-line DST for drugs AMK (1 mg/mL), LFX(1 mg/mL), LNZ(1 mg/mL), and MOX(1 mg/mL), 49 (69.0%) were found to be susceptible, 20 (28.2%) were mono-resistant to FQ, [2(2.8%) was resistant to only MOX and 18 (25.4%) to (MFX+LFX)]; 2 (2.8%) isolates were found to be resistant to (FQ+AMK) while no isolates were found mono-resistant to AMK or resistant to LNZ. Patterns of first- and second-line drugs are shown in Table 1.

### 3.5. Mutations in Genes Associated with First- and Second-Line Drugs Using WGS

A total of 65 isolates sequenced were analyzed for mutations conferring drug resistance in genes associated with first-line and second-line drug resistance (Table 2). Genomes sequences of each isolate were screened for mutations in genes conferring resistance to first-line anti-tuberculosis drugs viz; rpsL, rrs, gidB for STR; KatG, inhA, ahpA, fab, ndh for INH; rpoA, rpoB, rpoC for RIF; embABC for EMB and pncA for PZA.

Of the total 65 isolates, Lys43Arg (22/26; 84.6%) in rpsL, Ser315Thr (35/45; 77.8%) in katG, Ser531Leu (21/41; 51.2%) in rpoB, Met306Val (18/30; 60%) in embB, and Asp49Ala (5/10; 50%) in pncA was found to be predominantly present in genes known to confer drug resistance for first-line drugs STR, INH, RIF, EMB, and PZA, respectively.

All 65 sequenced genomes were also analyzed for mutations in genes conferring resistance to second-line anti-tuberculosis drugs viz gyrA, gyrB for FQ; rrl and rplC for linezolid; Rv0678, Rv2535c, Rv1979c and mmpl5 for clofazimine; alr, ddl, ald and cycA for cycloserine; rrs for AMK; Rv0678 and atpE for bedaquiline; thyA, ribD and folC for PAS; fgd, ddn, fbiA, fbiB and fbiC for delamanid.

In case of FQ, Asp94Gly (6/16; 37.5%) was found to be the predominant mutation in gyrA gene region, of which two genomes were also found to have mutation 1484G>T and 1401 A>G in rrs gene region conferring drug resistance to injectable class of drugs known to confer drug resistance. No mutations were found for genes associated with linezolid, clofazimine, delamanid, and bedaquiline.

Patterns of mutation resulting from WGS analysis and their association with phenotypic DST are shown in Table 2. Variant densities of each genome against *M. tuberculosis* H37Rv were generated using Blast Ring Image Generator BRIGv0.95 and is shown in Figure 2.

### 3.6. Phylogenetic Analysis and Identification of Lineages Based on cgMLST and SNP Barcoding

All 65 sequenced genomes were used along with 39 publicly available genomes (including 33 genomes representing lineage 1-8 of *M. tuberculosis* and 6 representatives from *M. tuberculosis* complex) to generate a phylogeny. The resulting tree showed the 65 isolates clustering with the publicly available lineage-defined genomes of *M. tuberculosis* (Figure 3). Lineage 2 (East-Asian) dominated the dataset with 41 (63.1%) genomes. Ten genomes (15.4%) were grouped in lineage 3 (East-African Indian) while 8 (12.3%) and 6 (9.2%) genomes were clustered with lineage 1 (Indo-Oceanic) and 4 (Euro-American), respectively. No genomes were clustered with lineage 5, 6, 7, and 8.

For all the 65 sequenced genomes, SNP barcoding was carried out on the basis of concatenated SNP’s. This barcoding analysis revealed that lineage 2 dominated the dataset, followed by lineage 3, 1, and 4, respectively. 34 out of 41 (82.9%) genomes of lineage 2 belonged to sub-lineage 2.2.1 (Beijing), while 7 (17.1%) genomes belonged to sub-lineage 2.2.1.2 (Beijing). Seven (87.5%) out of 8 lineage 1 genomes belonged to sub-lineage 1.1.3 (EAI6) while one genome (12.5%) was assigned to sub-lineage 1.1.2 (EAI5). of the total ten genomes of lineage 3, only one (10%) belonged to sub-lineage 3.1.2.1 (CAS2) while the remaining 9 (90%) were not assigned any sub-lineage. Lineage 4 consisted of total six genomes with 2 (33.3%), 1 (16.7%), 2 (33.3%) genomes belonging to sub-lineage 4.5 (H3; H4; T), 4.3 (LAM) and 4.1.1.1 (X2), respectively, whereas one genome (16.7%) was not assigned to any of the sub-lineage. The UPGMA tree generated using WGS SNP barcoding was congruent to the cgMLST tree based on lineage distribution (Figure 4).

### 3.7. Phylogenetic Analysis Based on PrincipalGenetic Group (PGG)

Based on the constitution of amino acids at loci 95 and 493, the PGG informative sites within the genes gyrA and katG, we found each isolate was designated to a PGG. Out of 65 sequenced genomes, 60 were clustered in PGG1, 4 in PGG2, and only 1 in PGG3. On comparing the data based on sub-lineage classification, all the genomes belonging to sub-lineage 1, 2, and 3 were clustered in PGG1 group, while sub-lineages of lineage 4 dominated PGG2 group. Only 1 genome from lineage 4 was assigned PGG3, which was a pre-XDR isolate. Assignment of PGG and sub-lineage classification for each isolate is shown in Appendix A.

## 4. Discussion

With increasing drug-resistant TB cases in India, it becomes pivotal to recognize the clonal expansion of lineages or clones contributing to drug resistance, specifically in geographical regions where drug resistance is suspected [35]. Northeastern states of India have higher rates of MDR-TB, around 32.7% of which Arunachal Pradesh region is known to have 78.8% of TB drug resistance [5,36]. Most of the land in Arunachal Pradesh is under forest area, and villages located in such impoverished forest zones have very inadequate or non-existent access to health services, including the hampered efforts of the national TB elimination program. Further, proximity to China, difficult terrains, and heavy rainfalls and landslides impact health services, leading to a high MDR rate. Nonetheless, host genetics also might be contributing to high resistance in this area, but that aspect was not studied in this work, and we are not making any conclusions on the genetic reasons for high MDR. Whole-genome sequencing (WGS) has become a standard for typing of *M. tuberculosis* isolates and is known to have higher resolution over MIRU-VNTR-based clustering [37]. In this study we planned to utilize the approach of WGS and SNP barcoding for typing and clustering of *M. tuberculosis* lineages circulating in the Arunachal Pradesh region of India. This study will help in defining the transmission of strains associated with drug resistance circulating in the Arunachal Pradesh region of India.

Our report for MDR-TB from Arunachal Pradesh is 56.8%, slightly lower than previously reported 79.3%. The variation may be due to a lower number of sample sizes in the previous study, results based on DNA-based line probe Assay rather than Bactec MGIT 960, which detects viable bacilli and incorrect denominators used while calculating drug resistance rates [36]. In our study, 80% of the samples were obtained from previously treated TB cases resulting in an increased rate of drug resistance. However, no reports of FQ resistance were reported from Arunachal Pradesh earlier.

There is mounting evidence that strain diversity plays a role in the transmission of disease [38]. In order to spot strain transmission in Arunachal Pradesh, we randomly selected cultures for WGS from each category of varying data sets of drug-resistant patterns and used gene by gene PGG, SNP barcoding, and core genome MLST (cgMLST) method to allocate strains in well-defined phylogenetic groupings (Figure 1). All these methods were used in various phylogenetic studies for standardized phylogenetic assignment of diseases and outbreak resolutions [13,39,40]. The PGG results also correlated with results of lineage grouping by SNP barcoding as 60 (92.3%) isolates belong to PGG group 1 (KatG Leu463Leu, gyrA Thr95Thr) of which 8 (13.3%) were Lineage 1 (Indo Oceanic), 41 (68.3%) Lineage 2 (East Asian) and 10 (16.7%) Lineage 3 (East-Arican Indian) and 1 (1.7%) Lineage 4 (Euro American). PGG group 2 (KatG Leu463Arg, gyrA Thr95Thr) and PGG group 3 (KatG Leu463Arg, gyrA Thr95Ser) included 4 (6.2%) and 1 (1.5%) strain belonging to lineage 4. Categorization of lineage as per the PGG group reported from other studies was consistent with our finding [31,40] (Appendix A).

This study provided for the first time a complete picture of TB phylogenetics across Arunachal Pradesh region based on cgMLST compared to phylogenetics that is based on SNP calling methods. The resulting cgMLST phylogenetic tree contained all reference lineages and four major *M. tuberculosis* lineages from our dataset and matched with results of SNP-based methods. None of the genomes belong to Lineage 5-8, and all were *M. tuberculosis sensu stricto* (Figure 1).

Using SNP barcoding method, we found 41 (63.1%) isolates grouped with lineage 2 (East Asian); 10 (15.4%) isolates with lineage 3 (Central Asian), 8 (12.3%) isolates with lineage 1, and 6 (9.2%) isolates to be lineage 4 (Appendix A). We found Lineage 2 (East Asian) as the predominant lineage (63.1%) circulating in Arunachal Pradesh among suspected drug-resistant TB cases. To gain further insight, we looked for SNP-based markers to differentiate sub-lineages. Among Lineage 2 only two clones circulating in Arunachal Pradesh were found viz: sub-lineage 2.2.1 (82.9%) and 2.2.1.2 (17.1%), respectively. Both these sub-lineages 2.2.1 and 2.2.1.2 belonged to modern Beijing clade, which was depicted by the presence of SNP markers at mutT2 codon Gly58Ala, ogt Gly12Gly specific to modern Beijing as reported in earlier studies [40,41]. Of a total of 6535 SNP’s, Beijing clone 2.2.1 was responsible for >80% of transmission and clusters using thresholds of up to 19 SNPs showing recent transmission of strains. Another clone of Beijing 2.2.1.2 showed clusters with SNP differences of 9 SNPs, showing ongoing transmission of the strains specifically associated with drug resistance. All these 7 (100.0%) isolates of clone 2.2.1.2 were multidrug-resistant, and 2 (28.6%) were resistant to FQ. Beijing sub-lineages 2.2.1 and 2.2.1.2 was reported from various parts of the world associated with outbreaks and drug resistance from Vietnam and Southern China [42,43,44,45,46,47]. We also observed Lineage 1 clone 1.1.3 with SNP differences of 101, showing this clone as endemic in Arunachal Pradesh for longer time periods. Lineage 1 is known to be associated with activation of long-term latent infection compared to that of Lineage 2 (modern Beijing) strains, which are known for more likely to progress to active disease in various host populations, more virulent, and thus highly transmissible [48,49]. A total of 6 isolates of lineage 4 were identified and were unclustered. Of Lineage 3, one clone including 9 (90%) isolates was found showing SNP differences of 18, also showing ongoing transmission. Out of 9 isolates, 3 (33.3%) were MDR-TB. One isolate (10%) of sub-lineage 3.1.2.1 was also found and was MDR-TB as well as resistant to FQ. New clades of lineage 3 were also reported to be circulating in the Assam region of India by Devi et al., which is consistent with our study [50].

The main strength of our study is that it highlights the higher rates of drug resistance in the Indian state of Arunachal Pradesh, which has a common border with China, and provides insights into the phylogenetic diversity of MDR-TB isolates from this state using cgMLST. These findings may have important implications in understanding the molecular epidemiology of DR-TB and for its control and prevention, particularly in the state of Arunachal Pradesh. However, our study has some limitations. The main limitation was that we could not include a control group from other states/regions of India in order to compare the strain diversity, specifically the Beijing sub-lineage 2.2.1 and 2.2.1.2, its association with drug resistance.

## 5. Conclusions

Our findings show dissemination of clusters of Beijing clones associated with drug resistance and regional spread may be emerging and aggressive. Approaches to contain Beijing strains (sub-lineage 2.2.1 and 2.2.1.2) may prevent transmission of these strains across other parts of India. Clonal expansion of these strains in Arunachal Pradesh in the future may lead to an outbreak of Beijing strains and underline the need for surveillance studies incorporating epidemiological information and a track of ongoing transmission to prevent drug-resistant TB outbreaks. We also found transmission of lineage 3 clade and the presence of Lineage 1 as endemic in Arunachal Pradesh. These findings may have important implications for control and prevention of TB in the northeastern part of India, Arunachal Pradesh.

## Figures and Tables

**Figure 1 genes-13-00263-f001:**
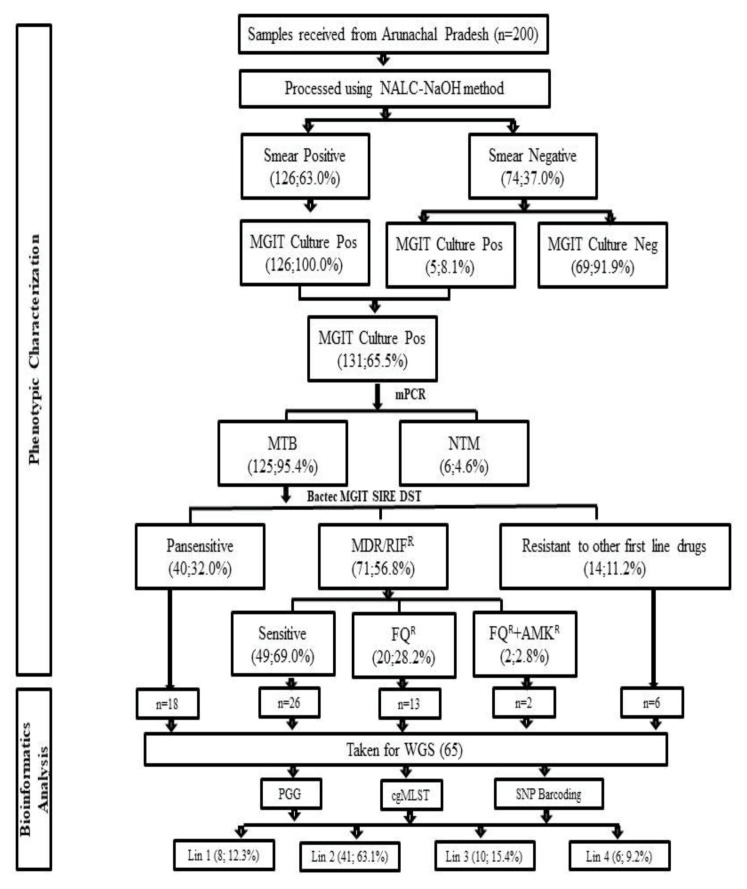
Workflow showing the phenotypic DST result and bioinformatics analysis of the isolates selected for WGS. FQ^R^: Fluoroquinolone Resistant; AMK^R^: Amikacin Resistant.

**Figure 2 genes-13-00263-f002:**
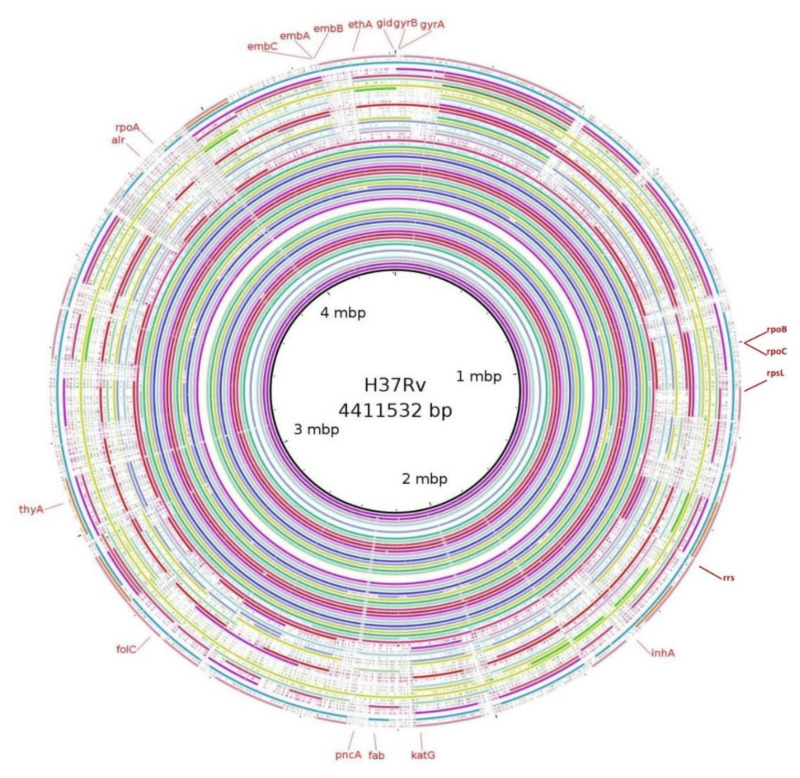
Variant densities of the 65 sequenced genomes against M. tuberculosis H37Rv using BRIG v0.95.

**Figure 3 genes-13-00263-f003:**
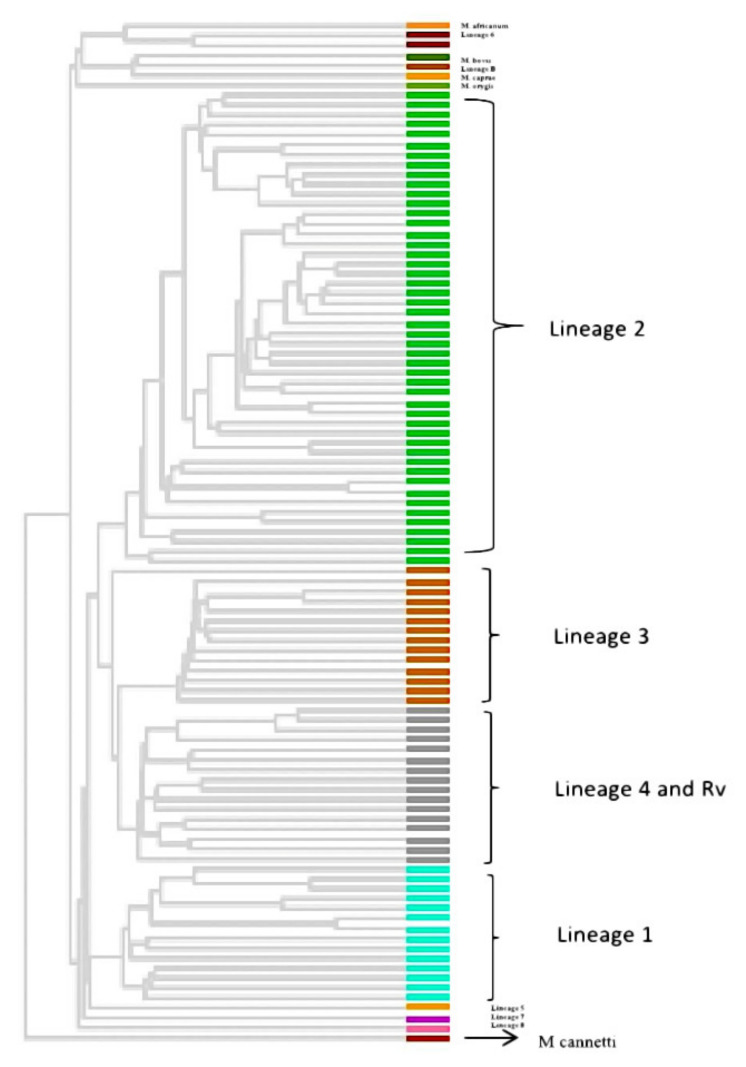
Phylogenetic analysis (UPGMA tree) based on cgMLST association of 104 genomes (65 isolates under this study and 33 lineage-defined genomes and 6 reference genomes of MTBC complex viz. *Mycobacterium africanum*, *Mycobacterium bovis*, *Mycobacterium caprae*, *Mycobacterium orygis*, *M. tuberculosis* and *Mycobacterium cannettii*). *Mycobacterium cannettii* was used to root the tree. Each lineage is shown with different colours.

**Figure 4 genes-13-00263-f004:**
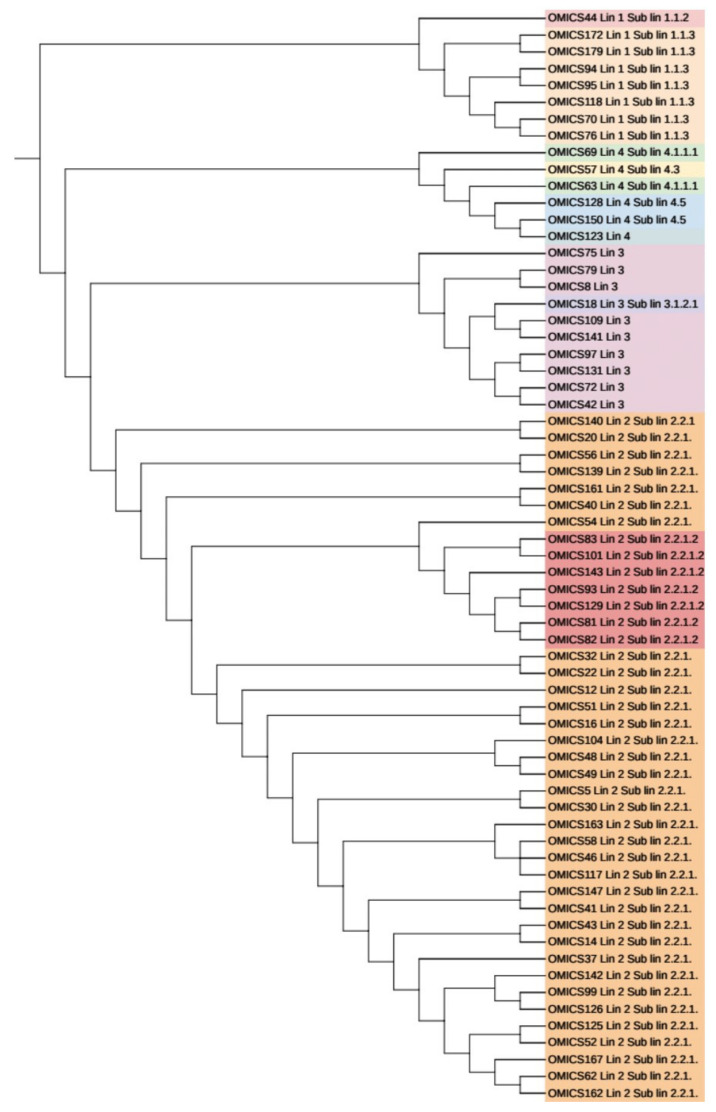
Phylogenetic analysis (UPGMA tree) of 65 sequenced genomes using SNP barcoding and assigned lineages as well as sub-lineages shown with different colors. Lineage 2 with sub-lineage 2.2.1 was found to be predominant in the state of Arunachal Pradesh.

**Table 1 genes-13-00263-t001:** Association of first-line drug-resistant patterns with second-line DST on Bactec MGIT-960.

SIRE Drug Susceptibility Pattern by MGIT 960						Second Line Drug Susceptibility Pattern by MGIT 960					
STR	INH	RIF	EMB	n	%	AMK	MFX	LFX	LNZ	n	%
R	R	R	R	37	29.6%	S	R	R	S	13	35.1%
						R	R	R	S	2	5.4%
						S	R	S	S	2	5.4%
						S	S	S	S	20	54.1%
S	R	R	S	15	12.0%	S	R	R	S	2	13.3%
						S	S	S	S	13	86.7%
S	R	R	R	5	4.0%	S	R	R	S	2	40.0%
						S	S	S	S	3	60.0%
R	R	R	S	9	7.2%	S	R	R	S	1	11.1%
						S	S	S	S	8	88.9%
S	S	R	S	3	2.4%	S	S	S	S	3	100.%
S	R	S	S	8	6.4%	−	−	−	−	−	−
S	S	R	R	1	0.8%	S	S	S	S	1	100%
R	S	R	S	1	0.8%	S	S	S	S	1	100%
R	R	S	R	2	1.6%	−	−	−	−	−	−
R	S	S	S	1	0.8%	−	−	−	−	−	−
S	S	S	R	2	1.6%	−	−	−	−	−	−
R	R	S	S	1	0.8%	−	−	−	−	−	−
S	S	S	S	40	32.0%	−	−	−	−	−	−
STR^S^ = 74 STR^R^ = 51	INH^S^ = 48 INH^R^ = 77	RIF^S^ = 54 RIF^R^ = 71	EMB^s^ = 78 EMB^R^ = 47	n = 125		AMK^S^ = 69 AMK^R^ =2	MFX^S^ = 49 MFX^R^ = 22	LFX^S^ =51 LFX^R^ = 20	LNZ^S^ = 71	n = 71	

STR^S^: Streptomycin susceptible; STR^R^: Streptomycin resistant; INH^S^: Isoniazid susceptible; INH^R^: Isoniazid resistant; RIF^S^: Rifampicin susceptible; RIF^R^: Rifampicin resistant; EMB^s^: Ethambutol susceptible; EMB^R^: Ethambutol resistant; AMK^S^: Amikacin susceptible; AMK^R^: Amikacin resistant; MFX^S^: Moxifloxacin susceptible; MFX^R^: Moxifloxacin resistant; LFX^S^: Levofloxacin susceptible; LFX^R^: Levofloxacin resistant; LNZ^S^: Linezolid susceptible.

**Table 2 genes-13-00263-t002:** Patterns of mutation resulting from WGS analysis and their association with phenotypic DST.

Drug	Gene	Phenotypic DST	Whole Genome Sequencing Results					
		MGIT Result	Mutation	No of Isolates		Lineages		
					Lin1	Lin2	Lin3	Lin4
RIF (n = 41)	*rpoB*	R	Ser531Leu	21/41 (51.2%)	4 (19%)	15 (71.4)	2)	−
		R	Leu511Pro Phe505Leu *	3/41 (7.3%)	−	3 (100%)	−	−
		R	Leu511Pro His526Gln	1/41 (2.4%)	−	1 (100%)	−	−
		R	Leu511Pro His526Gln Phe505Leu *	2/41 (4.9%)	−	2 (100%)	−	−
		R	Asp516Val	2/41 (4.9%)	−	1 (50%)	−	− (50%)
		R	Asp516Tyr	1/41 (2.4%)	−	1 (100%)	−	−
		R	His526Tyr	3 /41(7.3%)	1 (33.3%)	2 (66.7%)	−	−
		R	His526Asp	3/41 (7.3%)	-	3 (100%)	−	−
		R	Gln513Pro	1/41 (2.4%)	−		−	1 (100%)
	*rpoB* *rpoC*	R	Ser531Leu Ile561Val * Ile572Thr	1/41 (2.4%)	−	1 (100%)	−	−
	*rpoB* *rpoA*	R	Ser531Leu Gly112Ser	1/41 (2.4%)	1 (100%)	−	−	−
	*rpoB* *rpoA*	R	Ser531Leu Gly319Lys	1/41 (2.4%)	−	1 (100%)	−	−
	*rpoB* *rpoA*	R	Ser531Leu Val264Gly	1/41 (2.4%)	−	−	−	1 (100%)
	*rpoB*	S	Leu545Met *	1/41 (2.4%)	−	1 (100%)	−	−
INH (n = 45)	*KatG*	R	Ser315Thr	35/45 (77.8%)	3 (8.6%)	27 (77.1%)	3 (8.6%)	2 (5.8%)
		R	Ser450Leu	1/45 (2.2%)	1 (100%)	−	−	−
	*inhA*	R	Ser94Ala	2/45 (4.4%)	2 (100%)	−	−	−
	*ahp*	R	52C>T	2/45 (4.4%)	2 (100%)	−	−	−
	*Fab*	R	15C>T	1/45 (2.2%)		1 (100%)	−	−
	*Kat* *Fab*	R	Ser140Gly 15C>T	1/45 (2.2%)		1 (100%)	−	−
		R	Ser315Thr 17C>T	1/45 (2.2%)		1 (100%)	−	−
		R	Ser315Thr 15C>T	2/45 (4.4%)		2 (100%)	−	−
EMB (n = 30)	*embB*	R	Met306Val	18/30 (60.0%)	−	16 (88.8%)	−	2 (12.2%)
		R	Gly406Asp	1/30 (3.3%)	−	−	1 (100%)	−
		R	Met306Ile	4/30 (13.3%)	1 (25%)	3 (75%)		−
		R	Asp354Ala	1/30 (3.3%)	−	1 (100%)	−	−
		R	Met306Leu	1/30 (3.3%)	−	−	1 (100%)	−
	*embB*	R	Glu405Asp	1/30 (3.3%)	−	1 (100%)	−	−
		R	Gln853ProMet306Val	2/30 (6.7%)	−	2 (100%)	−	−
	*embA* *embA*	R	Met306Val 12C>T	2/30 (6.7%)	−	2 (100%)	−	−
STR (n = 26)	*rpsL*	R	Lys43Arg	22/26 (84.6%)	−	21 (95.4%)	1 (19.1%)	−
		R	Lys88Arg	2/26 (7.7%)	−	2 (100%)	−	−
	*rrs*	R	514 A>C	2/26 (7.7%)	−	2 (100%)	−	−
FQ (n = 16)	*gyrA*	R	Asp94Gly	6/16 (37.5%)	−	5 (83.3%)	1 (16.6%)	−
		R	Asp94Tyr	2/16 (12.5%)	−	2 (100%)	−	−
		R	Asp94Asn	2/16 (12.5%)	−	2 (100%)	−	−
		R	Asp94Ala	1/16 (6.3%)	−	1 (100%)	−	−
		R	Ala90Val	1/16 (6.3%)	−	−	−	− (100%)
		R	Asp94His	1/16 (6.3%)	−	1 (100%)	−	−
	*gyrB*	R	Ile486Leu	1/16 (6.3%)	−	1 (100%)	−	−
		R	Asp461His	1/16 (6.3%)	−	−	−	1 (100%)
		R	Ala504Val	1/16 (6.3%)	−	−	−	−
PZA (n = 10)	*pncA*	R	Asp49Ala	5/10 (50.0%)	−	5 (100%)	−	−
		R	Gly108Arg	2/10 (20.0%)	−	2 (100%)	−	−
		R	11A>G	2/10 (20.0%)	−	2 (100%)	−	−
		R	Asp136Tyr	1/10 (10.0%)	1 (100%)	−	−	−
AMK (n = 2)	*rrs*	R	1484 G>T	1/2 (50.0%)	−	1 (100%)	−	−
		R	1401 A>G	1/2 (50.0%)	−	1 (100%)	−	−
ETH (n = 8)	*inha*	NA	Ser94Ala	2/8 (25.0%)	2 (100%)	−	−	−
	*fab*	NA	15C>T	3/8 (37.5%)	−	3 (100%)	−	−
		NA	17G>T	1/8 (12.5%)	−	−	−	1 (100%)
	*ethA*	NA	886_886del	2/8(25.0%)	−	1 (50%)	−	1 (50%)
Cysr (n = 2)	*alr*	NA	Met343Thr	2/2 (100.0%)	−	2 (100%)	−	−
PAS (n = 2)	*thy*	NA	16C>T	1/2 (50.0%)	−	1 (100%)	−	−
	*folC*	NA	Ile43Thr	1/2 (50.0%)	−	1 (100%)	−	−

STR: Streptomycin; INH: Isoniazid; RIF: Rifampicin; EMB: Ethambutol; PZA: Pyrazinamide; ETH: Ethionamide; FQ: Fluoroquinolone; AMK: Amikacin; PAS: Para-aminosalicylic acid; Cysr.: Cycloserine; S: susceptible; R: resistant; * outside Rifampicin Resistance Determining Region (RRDR); n=number of isolates.

## Data Availability

Genomes of sixty-five *M. tuberculosis* isolates have been deposited in GenBank under BioProject accession no. PRJNA717132. Raw reads of all sixty-five *M. tuberculosis* have been made available in the Sequence Read Archive (SRA) under study number SRR331414 linked with BioProject number PRJNA717132.

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
