# Peer review of "Snapshot of Mycobacterium tuberculosis Phylogenetics from an Indian State of Arunachal Pradesh Bordering China"

_genes, 2022, doi:10.3390/genes13020263_

Round 1

Reviewer 1 Report

1.   Page 1: Introduction

The authors have cited Reference 5 to suggest high prevalence of 78.8% MDR-TB cases in Arunachal Pradesh (India). This is a very significantly high number of MDR TB case, considering the MDR-TB cases globally or in India. Reference number 5 by Singh et al. (2014) itself speculates about Reference 36 that “authors used incorrect denominators while calculating these drug resistance rates…”. Please clarify on these and provide appropriate references to support such high MDR-TB case.

2) Sentences require clarity at many places. The manuscript must be significantly proof-read and edited. Some of the sentences are:

Page 2: Different approaches were previously….signatures for phylogenetic categorization.

Page 3: As second line drug (SLD)….the reports obtained

Page 12: The PGG results also…Consistent with out finding

And elsewhere

  • Please correct Ziehl-Neelson to Ziehl-Neelsen
  • “Mycobacterium tuberculosis” must be in italics
  • Considering that the MDR-TB rate reported by the authors are very high as compared to the Global and Indian MDR-TB cases, the authors must provide reasonable evidence that the data is factually correct. In this regard, the authors must provide more clarity to the following:
  1. The depth of sequencing
  2. The exclusion and inclusion criteria of patients must be provided. Whether only NEW TB cases or Re-infected TB cases were also considered.
  • The explanations provided by authors for high MDR-TB case are “inhospitable topography and climate conditions”. While the geography and climate may have some impact, it is non-conceivable that these can be the dominant cause for such high MDR-TB case. The authors may look if the host genetics have any propensity for driving higher MDR-TB case in Arunachal Pradesh. Please clarify. This must be clearly explained in discussion.
  1. Please provide reference for the statement “strain diversity plays a role in transmission of disease”.
  • Supplementary Table 1 is not provided.
  • Phrases such as “First kind of study..” “Study provided for first time..” must be avoided.
  • Figures are of poor resolution and hence it is difficult to comprehend with the results in text.

Author Response

¸

Reviewer’s comments (Reviewer 1)

Author’s response

1.

Page 1: Introduction

The authors have cited Reference 5 to suggest a high prevalence of 78.8% MDR-TB cases in Arunachal Pradesh (India). This is a very significantly high number of MDR TB cases, considering the MDR-TB cases globally or in India. Reference number 5 by Singh et al. (2014) itself speculates about Reference 36 that “authors used incorrect denominators while calculating these drug resistance rates…”. Please clarify on these and provide appropriate references to support such a high MDR-TB case.

The keen observation of the reviewer is appreciated. The editorial questioned te denominator used by the authors but not the high resistance rate in these states. For that reason, the editorial summarises in the end as “Data on drug resistance and various genotypes circulating in North-Eastern States of India is scarce. Hence this study provides useful important information from this region. The alarming drug resistance rates reported here, even if adjusted with correct denominators, are bothering and steps to contain the spread of drug resistant TB to neighbouring areas are urgently required.”

2.

Sentences require clarity at many places. The manuscript must be significantly proof-read and edited. Some of the sentences are:

Page 2: Different approaches were previously….signatures for phylogenetic categorization.

Page 3: As second line drug (SLD)….the reports obtained

Page 12: The PGG results also…Consistent with out finding

And elsewhere

We thank the reviewer for the suggestion and apologise for the errors. We have done proofreading and language has been improvised throughout the manuscript.

3.

Please correct Ziehl-Neelson to Ziehl-Neelsen

We thank the reviewer for the suggestion and apologise for the errors. Now corrected throughout the manuscript

4.

“Mycobacterium tuberculosis” must be in italics

Corrected throughout the manuscript

5.

Considering that the MDR-TB rate reported by the authors are very high as compared to the Global and Indian MDR-TB cases, the authors must provide reasonable evidence that the data is factually correct. In this regard, the authors must provide more clarity to the following:

  1. The depth of sequencing
  2. The exclusion and inclusion criteria of patients must be provided. Whether only NEW TB cases or Re-infected TB cases were also considered.

We thank the reviewer for pointing us towards the high MDR-TB finding in our manuscript.

We re-evaluated the patient's history and found a total of 200 patients included in the study. 22 (11%) patients were suspected of MDR-TB and 160 (80%) patients were previously treated cases. Of 22 (11%) suspected MDR-TB cases 7 (31.8%) were culture positive for MTB of which 4 were MDR/RR TB by phenotypic DST and of 160 previously treated cases 118 (73.75%) were culture positive for MTB of which 66 were MDR/RR TB cases by phenotypic DST.

Thus, of 125 MTB cultures,  5 (4%) were new MDR/RR TB cases and 66 (52.8%) were previously treated MDR-TB/RR-TB cases.

Mutations with read depth above ten reads were considered true mutations. (included in the Methods)

6.

The explanations provided by authors for high MDR-TB cases are “inhospitable topography and climate conditions”. While the geography and climate may have some impact, it is non-conceivable that these can be the dominant cause for such high MDR-TB case. The authors may look if the host genetics have any propensity for driving higher MDR-TB case in Arunachal Pradesh. Please clarify. This must be clearly explained in discussion.

We thank the reviewer for the important suggestion and we agree with that. Most of the land in Arunachal Pradesh is under forest area and villages located in such impoverished forest zones have very inadequate or non-existent access to health services including the hampered efforts of national TB elimination programme. Further difficult terrains and heavy rainfalls and landslides, impact on health services. Nonetheless, as pointed out by the reviewer, host genetics also might be contributing to high resistance, but since we have not study that aspect, we are confident to mention it with confidence, but the discussion has been modified accordingly.

7.

Please provide reference for the statement “strain diversity plays a role in transmission of disease”.

Chizimu et al 2022 (Ref. 38). Genetic Diversity and Transmission of Multidrug-Resistant Mycobacterium tuberculosis strains in Lusaka, Zambia.

https://www.ijidonline.com/article/S1201-9712(21)00826-2/fulltext

8.

Supplementary Table 1 is not provided.

Included.

9.

Phrases such as “First kind of study..” “Study provided for first time..” must be avoided.

We have excluded these terms from the Manuscript.

10.

Figures are of poor resolution and hence it is difficult to comprehend with the results in text.

Figure resolutions are increased to 600dpi.

Reviewer 2 Report

With increasing drug-resistant TB cases, it is worth recognizing
the clonal expansion of lineages or clones contributing to drug resistance specifically in geographical regions where drug resistance is suspected. This is what  S. Rashmi Mudliar et al investigated in  Northeastern states of India which had higher rates of MDR-TB.
The manuscript is well structured. However, the authors might consider the following comment for improvement 
1-Page 2, method, sample collection: Please add the time (period) of sample collection
2-Page 5: “Of total 200 patients included in the study, 91 (45.5%) were male and 109 (54.5%)
females” no need to add female percentage since with the male's, it could be deduced.
3-What is missed are the strength and limitations of the study, please add 
4--Other comment: Please add the lines number  to facilitate me indicate the line where my comment referred 

Author Response

S. No.

Reviewer’s comments (Reviewer 2)

Author’s response

1.

Page 2, method, sample collection: Please add the time (period) of sample collection

Samples were collected from September 2019 to September 2021.

2.

Page 5: “Of total 200 patients included in the study, 91 (45.5%) were male and 109 (54.5%)
females” no need to add female percentage since with the male's, it could be deduced.

We have corrected as mentioned.

3.

What is missed are the strength and limitations of the study, please add 

This study provides insights into the phylogenetic diversity of MDR-TB isolates from Arunachal Pradesh using cg MLST. We concluded that dissemination of two Beijing clones associated with drug resistance in Arunachal Pradesh belonged to sublineage 2.2.1 and 2.2.1.2.. These findings may have important implications for the control and prevention of drug-resistant TB in the north-eastern part of India, Arunachal Pradesh.

 The limitation of this study is we have not included a control group from other regions of India in order to compare the strain diversity specifically Beijing clones associated with drug resistance.